# Diminishing the Gender-Related Disparity in Survival among Chemotherapy Pre-Treated Patients after Radical Cystectomy—A Multicenter Observational Study

**DOI:** 10.3390/jcm12041260

**Published:** 2023-02-05

**Authors:** Krystian Kaczmarek, Artur Lemiński, Bartosz Małkiewicz, Adam Gurwin, Janusz Lisiński, Marcin Słojewski

**Affiliations:** 1Department of Urology and Urological Oncology, Pomeranian Medical University, Powstańców Wielkopolskich 72, 70-111 Szczecin, Poland; 2Department of Minimally Invasive and Robotic Urology, University Center of Excellence in Urology, Wroclaw Medical University, Borowska 213, 50-556 Wrocław, Poland

**Keywords:** bladder cancer, gender disparity, neoadjuvant therapy, radical cystectomy

## Abstract

There is a well-documented problem of inferior outcome of muscle-invasive bladder cancer (MIBC) after radical cystectomy (RC) in women. However, previous studies were conducted before neoadjuvant chemotherapy (NAC) was widely adopted to multidisciplinary management of MIBC. In our study, we assessed the gender-related difference in survival between patients who received NAC and those who underwent upfront RC, in two academic centers. This non-randomized, clinical follow-up study enrolled 1238 consecutive patients, out of whom 253 received NAC. We analyzed survival outcome of RC according to gender between NAC and non-NAC subgroups. We found that female gender was associated with inferior overall survival (OS), compared to males (HR, 1.234; 95%CI 1.046–1.447; *p* = 0.013) in the overall cohort and in non-NAC patients with ≥pT2 disease (HR, 1.220 95%CI 1.009–1.477; *p* = 0.041). However, no gender-specific difference was observed in patients exposed to NAC. The 5-year OS in NAC-exposed women in ≤pT1 and ≥pT2 disease, was 69.333% 95%CI (46.401–92.265) and 36.535% (13.134–59.936) respectively, compared to men 77.727% 95%CI (65.952–89.502) and 39.122% 95%CI (29.162–49.082), respectively. The receipt of NAC not only provides downstaging and prolongs patients’ survival after radical treatment of MIBC but may also help to diminish the gender specific disparity.

## 1. Introduction

According to population-based studies, women have better survival rates, compared to men, after diagnosis of most cancers [1]. However, no such phenomenon is observed in bladder cancer (BC). The risk of BC progression and mortality is higher in women than in men [2]. In contrast, men are more likely to develop BC than women [1]. This gap in oncological outcomes may be partially attributed to more advanced stage at presentation in women [3]. The initially higher T-stage is mainly explained by a longer delay in the diagnosis of BC in women than in men, whereby a urinary tract infection (UTI) might be regarded as the culprit of a misdiagnosis of BC due to association with hematuria and higher prevalence among women. Although UTIs are more common in women <40 years old, a limited assessment of a gross hematuria is still seen in older women [4]. The gender-related survival disparity is also observed in stage-to-stage comparisons between women and men and is particularly pronounced in advanced stages of BC; hence, higher presenting stage in women cannot fully clarify this phenomenon [3]. Moreover, differences in gender-specific survival persist after radical cystectomy (RC). This emphasizes the complexity of this issue and the potential impact of other factors, such as treatment availability and patterns, tobacco smoking, chemical exposure, and hormonal status [5]. An association of gender with long-term outcomes of RC has been extensively analyzed before the widespread adoption of neoadjuvant chemotherapy (NAC) [6,7]. Administration of NAC is the current standard of care for patients with muscle-invasive bladder cancer (MIBC) eligible for cisplatin therapy and provides up to 50% of downstaging to <ypT2N0, with complete remissions achievable in one-third of patients [8,9]. Moreover, NAC improves overall survival (OS) by 5–8% at 5 years, equivalent to a number needed to treat of 12.5 [10]. Besides NAC, currently there is an increasing number of promising precisely targeted agents which are assessing for use in neoadjuvant setting in BC. These agents might be use alone or in combination with traditional chemotherapy [11]. However, cisplatin-based combination chemotherapy remains a recommended treatment before RC. Nevertheless, data regarding the gender inequality in survival after RC in the NAC era are limited. Therefore, in this evolving landscape of treatment for MIBC, we decided to conduct a study which answers a question of whether the gender-related disparity in survival in patients undergoing upfront RC would still be observed after exposure to NAC.

In the present study, we hypothesized that the administration of NAC not only improves mid-term outcome after RC, but may as well diminish the survival disadvantage among women after RC. Hence, the objective of our study was to reassess the differences in gender-specific survival between chemotherapy pre-treated and chemotherapy-naïve patients undergoing RC in the NAC era. 

## 2. Material and Methods

This non-randomized clinical follow-up study was exempt from further review by the Institutional Review Board (Bioethical Committee) of the Pomeranian Medical University, Szczecin, Poland (protocol number KB. 006.102.2022/Z-9521) and was conducted according to the regulations set forth by the Declaration of Helsinki. Consent for research participation was routinely obtained from all patients involved, namely, for the use of their anonymized treatment data collected during hospitalization. Consecutive patients who underwent RC and pelvic lymphadenectomy due to MIBC at two university centers—the Department of Urology and Urological Oncology of the Pomeranian Medical University, Szczecin, Poland, and the Department of Minimally Invasive and Robotic Urology, University Center of Excellence in Urology of Wroclaw Medical University, Poland—were enrolled in the study. Patients were treated between 1991–2021 and 2003–2021, in first and second participating department, respectively. Patients with metastatic disease, those who underwent cystectomy for palliative indications, those who underwent partial bladder resections, patients with a history of pelvic radiotherapy, and those with non-urothelial pathology were excluded from analysis. In total, 99 patients were excluded, and the data of 1238 patients were utilized for statistical analyses. The analyzed cohort was divided into two groups according to NAC administration. NAC was administered to 253 patients until the end of 2021. The remaining study population underwent upfront RC with eventual adjuvant treatment depending on the final pathologic stage (non-NAC group, *n* = 985). The NAC cisplatin-based chemotherapy was offered patients who meet all eligibility criteria, including Eastern Cooperative Oncology Group Performance Status (ECOG PS) 0–1, glomerular filtration rate (GFR) > 60 mL/min, audiometric hearing loss grade < 2, peripheral neuropathy grade < 2, and function of the heart according to the New York Heart Association Functional Classification < III. If patent did not fulfill all those criteria and ECOG PS was at least 2 and GFR was between 30–60 mL/min, then carboplatin-based chemotherapy was considered. However, if GFR was <60 mL/min and ECOG PS < 2 and the patient had adequate bone marrow reserve, then taxane-based chemotherapy was offered after individual assessment by oncological team. No patient was exposed to immuno-oncology therapy before RC. To identify differences between genders, we evaluated the association between gender and long-term oncological outcome in both groups according to the pathological stage (Figure 1). Data were reviewed for internal consistency. Descriptive statistics including mean ± standard deviation (SD) and median (interquartile range, IQR) were applied for normally distributed and skewed data, respectively. Single variables were compared using an independent t-test (parametric variables) and a chi-square test (non-parametric variables). If the Cochran’s assumptions for chi-square test were not met, the Fisher–Freeman–Halton test was applied. OS probabilities over time were presented using Kaplan–Meier survival estimates and univariate Cox models. The survival curves of different groups were compared using the log-rank test. Multivariable Cox proportional hazard models were applied to examine the impact of prognostic factors on survival, including age at the time of surgery, gender, the severity of comorbidities reflected by the American Society of Anesthesiologists (ASA) score, pathological T stage, cancer grade, and pathological lymph node status. To test for independence between residuals and time, an assessment of the proportional hazard assumption of final multivariable models was performed using scaled Schoenfeld residuals with time. The results of Cox proportional hazard models were presented as hazard ratios (HR) with their 95% confidence intervals (CIs). Additionally, because patients were not randomly assigned to administration of NAC, we performed propensity scores analyses. Logistic regression was used to calculate propensity scores to estimate the predictive probabilities of receiving NAC. We considered *p* value < 0.05 as statistically significant and all *p* values were two-sided. All tests were performed using Statistica software, version 13.5 (StatSoft, Inc., Tulsa, OK, USA) and R (version 4.2.2) and RStudio (version 2022.12.0) with R packages *survival*, *survminer*, *drylr.*

## 3. Results

Out of 1238 patients included in the study, 253 (20.4%) received NAC. The median follow-up time was 23.467 months (IQR, 8.800–52.233), and there were no significant differences between genders (women, 21.333 months [IQR, 8.783–50.667] vs. men, 23.967 months [IQR, 8.800–52.233]). Women were generally older at the time of surgery and had higher pathological cancer stage than men. Statistically, greater proportion of women had extravesical disease (65.98% and 57.14% respectively; *p* = 0.007). However, there were no significant differences in lymph node metastasis, tumor grade, and ASA score distributions (Table 1). 

No significant difference between women and men was found regarding exposure to NAC (women 20.08% vs. men 20.52%), chemotherapy regimen, and number of NAC cycles administered (Table 1). NAC provided disease downstaging to <ypT2N0 in 86 (33.99%) patients, including complete responses (CR) in 43 (17.00%; Table 2). No significant differences between women and men were observed in response rates to NAC. Complete remissions were observed in 22.45% of women and 14.71% of men (*p* = 0.187), whereas partial responses (<ypT2N0) were noted in 34.69% of women and 30.39% of men (*p* = 0.560). However, there were significant differences in response to NAC between participating centers. In the Department of Urology and Urological Oncology in Szczecin, CR was achieved in 22% of patients, whereas the rate of complete remissions in the Department of Minimally Invasive and Robotic Urology in Wrocław reached only 12% (*p* = 0.043). Correspondingly, significant difference was observed in partial response rates (40.0% and 25.49% respectively, *p* = 0.015). Further heterogeneity was observed regarding preferred cytotoxic regimens. In Szczecin, 59 (59%) patients received dose-dense methotrexate, vinblastine, doxorubicin, and cisplatin (ddMVAC) regimen, as compared to 58 (37.9%) patients who were exposed to NAC in Wroclaw (*p* < 0.001). Furthermore, fewer patients received an optimal number of NAC cycles (≥3) in Wroclaw, as compared to Szczecin (55.6% versus 72.0%, *p* = 0.008). 

Kaplan–Meier survival curves and Cox regression analyses were performed in the entire cohort and revealed that OS observed in women was inferior to that in men (HR, 1.234; 95% CI, 1.046–1.447; *p* = 0.013). The 5-year OS for women and men was 32.906% (95% CI, 26.215–39.597) and 41.819% (95% CI, 38.377–45.261), respectively. After stratification by pT stage, inferior OS was observed in women with ≥pT2 disease, as compared to men (HR, 1.218; 95%CI, 1.017–1.458; *p* = 0.032). The 5-year OS for women and men with ≥pT2 disease was 24.926% (95% CI, 18.267–31.585) and 32.596% (95% CI, 28.821–36.371), respectively. The difference was consistently observed in a subset of patients with ≥pT2 disease who did not receive NAC, with a 5-year OS of 25.511% (95% CI, 18.688–32.334) in women as opposed to 32.596% (95% CI, 28.821–36.371) in men (HR, 1.220; 95%CI, 1.009–1.477; *p* = 0.041). However, no gender-specific difference was observed across stage groups in patients who received NAC. The 5-year OS in the NAC group for women in ≤ypT1 and ≥ypT2 disease was 69.333% (95% CI, 46.401–92.265) and 36.535% (95% CI, 13.134–59.936), respectively as opposed to 77.727% (95% CI, 65.952–89.502) and 39.122% (95% CI, 29.162–49.082), in men (Figure 2). These findings were validated using a competing risk regression model. In multivariable analysis, the difference in OS was observed in the non-NAC ≥pT2 group (HR, 1.229; 95%CI, 1.013–1.492; *p* = 0.036). However, administration of NAC diminished gender-specific disparities in OS after RC also among patients who did not respond to neoadjuvant therapy (HR, 1.136; 95%CI, 0.630–2.046; *p* = 0.672; Table 3; Appendix A). These results were confirmed in propensity scores analysis (Appendix A).

## 4. Discussion

Gender-related survival differences in BC are well documented and more pronounced among patients undergoing RC [5,12,13,14]. These findings are surprising given that women generally have better disease-specific survival than men for most cancers affecting both sexes [15,16,17]. This phenomenon has not been fully elucidated, with current hypotheses including longer delay from presentation with hematuria to diagnosis in women, anatomic and hormonal differences, higher stage at presentation, and variation in tumor biology [4,11,18,19,20]. However, once diagnosed with MIBC, data on sex-specific responses to different treatment regimens and the influence on oncological outcomes is limited. 

In our study, women undergoing RC due to MIBC showed inferior OS when compared with men, which is in line with previous reports [5,14]. However, the difference in our cohort was more significant. The HR in the overall cohort was 1.234 (95% CI, 1.046–1.447). According to Krimphove et al., who used data from the National Cancer Database, women presenting with MIBC accounted for only 4% and were slightly more likely to die than men (HR, 1.04; 95% CI, 1.00–1.07) [21]. The difference was even less pronounced in other studies [2,22]. Variations in observed outcomes between the studies may be attributed to the differences in sample size, period of treatment, and study design. Our cohort was more homogeneous than cohorts in previous studies as we excluded patients with pathological variants of BC other than urothelial carcinoma and included patients who underwent RC at academic centers. Moreover, the differences between gender may be related to the increasing prevalence of tobacco-related BC among women in Poland. The highest exposure to smoking, which reached 50%, was reported in generation of Polish women born between 1940 and 1960 [23]. Therefore, considering the latency time and cohort effect, the incidence and mortality due to BC in women showed an increasing trend, which may explain the higher disproportion in OS observed between genders in our study. 

In the subgroup analysis of patients with ≥pT2 disease sex-related differences observed for all cohort persisted, with unadjusted and adjusted HR of 1.218 (95% CI, 1.017–1.458) and 1.219 (95% CI, 1.015–1.463), respectively. After further stratification of this group according to NAC exposure, the survival gap was evident in chemotherapy-naïve patients (HR, 1.229; 95% CI, 1.013–1.492), but was no longer found in patients who received NAC, in whom OS of women was comparable to that of men (HR, 1.136; 95% CI, 0.630–2.046). Equivalent results were presented in meta-analysis by Kimura et al. who analyzed the association between gender and oncologic outcomes in the NAC pretreated cohort. In their study, gender was not associated with overall mortality. Similarly, no correlation with CR or partial response was found. Of note, women undergoing NAC and RC were less likely to have tumor upstaging at the time of surgery than men [24]. 

Diminishment of gender-related OS gap in NAC pre-treated patients after RC may be partially explained at molecular level. According to the Cancer Genome Atlas, MIBC is divided into two main subgroups: luminal and basal-squamous. Each type is associated with distinct histopathological features, prognosis, and treatment implications [25]. Robertson et al. discovered that BC in women was mostly categorized as the basal-squamous subtype, which expressed high levels of CTLA4 and CD274 (PD-L1) immunomarkers and responded well to immune therapy [26]. Moreover, Seiler et al. highlighted that this molecular subtype showed greater response rate and survival improvement after NAC-RC, compared to other subtypes and cystectomy alone. Given higher proportion of basal-squamous subtype in women, we hypothesize NAC may yield better responses in female patients [27].

Another potential reason for comparable survival outcomes observed between sexes after NAC exposure may be related to difference in bladder wall thickness. The bladder wall is thinner in women than in men, which may facilitate more thorough transurethral resection of bladder tumor (TURBT) [28]. James et al. showed the association between maximal TURBT and improved survival and oncologic outcomes in a cohort of patients with MIBC. NAC pretreated patients who underwent maximal TURBT were more likely to achieve a complete pathologic response (odds ratio 3.17; 95% CI, 1.02–9.83) [29]. Similar results were reported by other authors [30]. 

Several NAC regimens were analyzed in this study. A significant proportion of patients were treated with ddMVAC regimen (46.25%), which may have affected our results. Haines et al. analyzed the association between gender and outcomes in patients with metastatic urothelial carcinoma and revealed a significantly longer survival in women than in men treated with MVAC (methotrexate, vinblastine, doxorubicin, and cisplatin), whereas no gender-related differences were observed in patients treated with gemcitabine and cisplatin regimens. However, these findings should be interpreted with caution given the small sample size of female patients treated with MVAC [31]. Therefore, an improved female response to ddMVAC is a hypothesis rather than a firm conclusion [9]. However, gender-related differences in response to particular regimens are not implausible.

Our present study has limitations, and the results should be interpreted within the limitations of the observational study design. Firstly, we were unable to analyze unmeasured confounding factors, such as socioeconomic status, comorbidities using the Charlson Comorbidity Index, smoking status, clinical stage, and adjuvant chemotherapy, which may have influenced survival. Hence, we also could not have provided a rate of clinical understaging in patients who underwent upfront RC. Secondly, there were significant disparities between participating centers regarding chemotherapy regimens, number of administered cycles, and response to neoadjuvant treatment. A plausible explanation of this phenomenon may include different treatment periods. Patients from the Department of Urology in Szczecin were treated more recently, when the role of NAC in the treatment of MIBC was firmly established, whereas in Wroclaw, the analysis, spanning over a period of 30 years, included the early era of NAC [32]. This discrepancy may be one of reasons why more patients in Wroclaw received less optimal regimens. Thirdly, as data regarding cause of death was not always available, our survival analysis was limited to OS, which may have introduced additional bias due to concurrent mortality, potentially significant in this population. Lastly, we are aware that currently there are ongoing studies, investigating novel therapeutic opportunities in neoadjuvant setting in MIBC. Preliminary results of these trials suggested that using agents such as checkpoint inhibitors before RC has clear potential advantages, due to their acceptable tolerance and high efficacy. Hence, soon, we might see changes in the landscape of treatment of BC [33,34]. Moreover, currently observing gender-related disparities among patients who are not exposing to neoadjuwant therapy might not be observed as more patients will be eligible to multidisciplinary approach. Thus, only palliative cases will be treated with upfront cystectomy.

Despite these limitations, we believe that this study provides a reliable analysis of gender-specific outcomes of early multidisciplinary management of MIBC. One of the strengths of the present study is the novelty of results addressing diminishing gender-specific disparities in survival in patients regardless of response status to NAC. In previous reports addressing sex-specific disparities after NAC administration, a reduction of survival gap was only observed in patients with pT4 disease [35,36]. Our results highlighted that women who have no pathological response to NAC may benefit from multidisciplinary treatment. Administration of NAC may help to mediate influence of potential sex-related factors, such as delayed diagnosis, anatomical differences, higher stage at the time of diagnosis, or altered tumor biology, which likely contribute to differences in oncologic outcomes.

## 5. Conclusions

In summary, our results highlight that the administration of neoadjuvant chemotherapy helps decrease the gender-related survival gap after radical cystectomy. Thus, we believe that female patients may benefit from gender-specific counseling regarding perioperative therapeutic strategies and personalized medicine approaches to greater extent than male patients. However, lack of gender-associated disparities in the survival of patients with muscle invasive bladder cancer after multidisciplinary treatment cannot be entirely proven based on the limitations of the study and further large, prospective studies are needed. Furthermore, in the light of an ongoing era of immunotherapy in advanced bladder cancer, the gender-related disparities should be studied simultaneously with the efficacy of new agents.

## Figures and Tables

**Figure 1 jcm-12-01260-f001:**
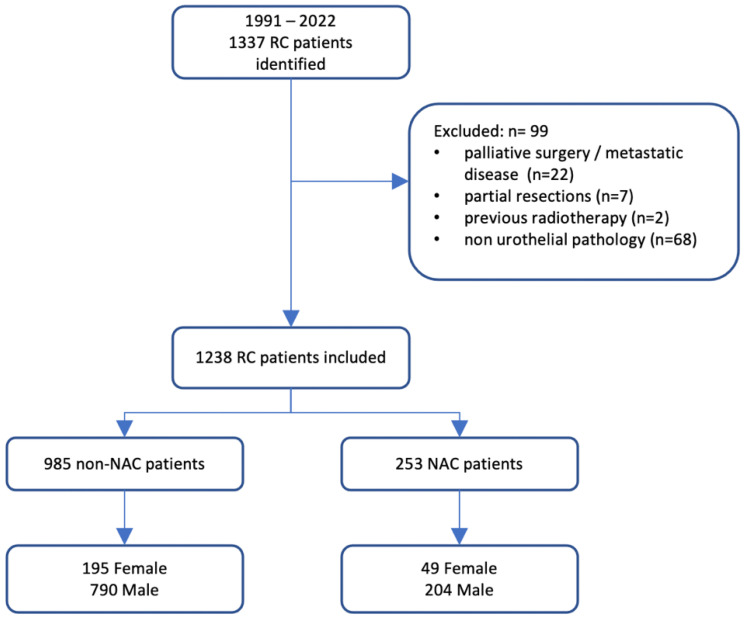
Flowchart of the study. NAC: neoadjuvant chemotherapy; RC: radical cystectomy.

**Figure 2 jcm-12-01260-f002:**
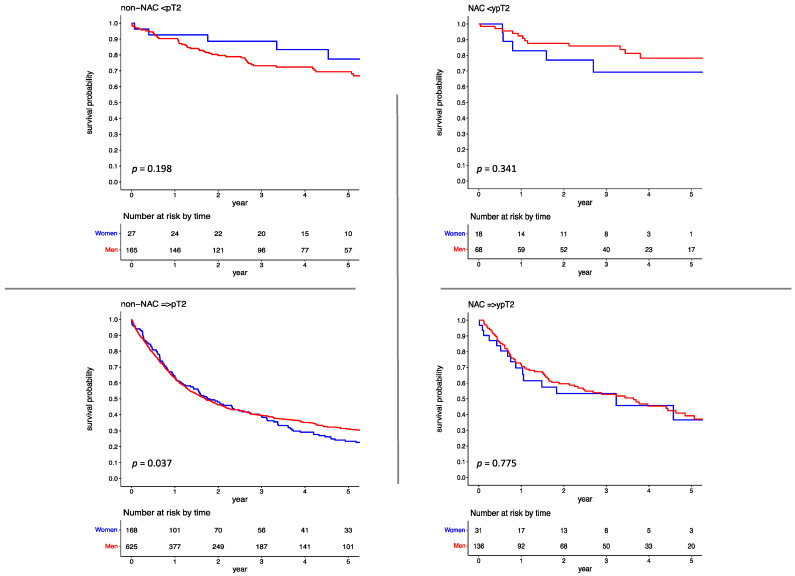
Kaplan–Meier analysis of overall survival in chemotherapy-naïve and chemotherapy pre-treated patients who underwent radical cystectomy stratified according to tumor stage.

**Table 1 jcm-12-01260-t001:** Baseline patients’ characteristics according to gender.

Variable	Female	Male	*p* Value
Totals, No. (%)	244 (19.71)	994 (80.29)	
Age, years			0.006
Mean	66.709	64.79	
SD	9.88	8.64	
ASA score, No. (%)			0.082
1	17 (6.97)	99 (9.96)	
2	169 (69.26)	611 (61.47)	
3	58 (23.77)	277 (27.87)	
4	0 (0.00)	7 (0.70)	
Pathological T stage, No. (%)			0.007
pT0	23 (9.43)	80 (8.05)	
pTis/Ta/T1	22 (9.02)	153 (15.39)	
pT2	38 (15.57)	193 (19.42)	
pT3	95 (38.93)	288 (28.97)	
pT4	66 (27.05)	280 (28.17)	
Pathological N stage, No. (%)			0.319
pN0	151 (61.89)	649 (65.29)	
pN+	93 (38.11)	345 (34.71)	
Cancer grade, No. (%)			0.213
Low grade	12 (4.92)	71 (7.14)	
High grade	232 (95.08)	923 (92.86)	
Neoadjuvant chemotherapy, No. (%)			0.878
No	195 (79.92)	790 (79.48)	
Yes	49 (20.08)	204 (20.52)	
Chemotherapy regimen, No. (%)			0.515
ddMVAC	21 (8.61)	96 (9.66)	
Gemcitabine-cisplatin	27 (11.07)	92 (9.26)	
Gemcitabine-carboplatin	0 (0.00)	5 (0.50)	
Gemcitabine-paclitaxel	1 (0.41)	11 (1.11)	
Cycles of chemotherapy, No, (%)			0.894
<3	19 (7.79)	77 (7.75)	
≥3	30 (12.30)	127 (12.78)	
Department, No. (%)			0.044
Szczecin	117 (47.95)	406 (40.85)	
Wrocław	127 (52.05)	588 (59.15)	
Follow-up, months			0.746
media	21.333	23.967	
IQ	8.783–50.667	8.800–52.233	

ASA score: American Society of Anesthesiologists score; IQR: interquartile range; SD standard deviation.

**Table 2 jcm-12-01260-t002:** Baseline patients’ characteristics according to expose to neoadjuvant chemotherapy.

Variable	non-NAC	NAC	*p* Value
Totals, No. (%)	985 (79.56)	253 (20.44)	
Age, years			0.431
Mean	65.27	64.77	
SD	9.13	8.11	
Gender, No. (%)			0.878
Female	195 (19.80)	49 (19.37)	
Male	790 (80.20)	204 (80.63)	
ASA score, No. (%)			0.084
1	96 (9.75)	20 (7.91)	
2	633 (64.26)	147 (58.10)	
3	251 (25.48)	84 (33.20)	
4	5 (0.51)	2 (0.79)	
Pathological T stage, No. (%)			<0.001
pT0	60 (6.09)	43 (17.00)	
pTis/Ta/T1	132 (13.40)	43 (17.00)	
pT2	177 (17.97)	54 (21.34)	
pT3	330 (33.50)	53 (20.95)	
pT4	286 (29.04)	60 (23.72)	
Pathological N stage, No. (%)			0.161
pN0	627 (63.65)	173 (68.38)	
pN+	358 (36.35)	80 (31.62)	
Cancer grade, No. (%)			<0.001
Low grade	81 (8.22)	2 (0.79)	
High grade	904 (91.78)	241 (99.21)	
Chemotherapy regimen, No. (%)			
ddMVAC	n/a	117 (9.44)	
Gemcitabine-cisplatin	n/a	119 (9.60)	
Gemcitabine-carboplatin	n/a	5 (0.40)	
Gemcitabine-paclitaxel	n/a	12 (0.97)	
Cycles of chemotherapy, No, (%)			
<3	n/a	96 (7.75)	
≥3	n/a	157 (12.68)	
Department, No. (%)			0.326
Szczecin	423 (42.94)	100 (39.53)	
Wrocław	562 (57.06)	153 (60.47)	
Follow-up, months			0.155
median	21.77	29.40	
IQR	8.00–53.27	10.43–49.57	

ASA score: American Society of Anesthesiologists score; IQR: interquartile range; SD standard deviation.

**Table 3 jcm-12-01260-t003:** Impact of female sex on overall survival in univariable, multivariable and propensity-weighted regression analysis stratified by pT stage and receipt of neoadjuvant chemotherapy.

	UNIVARIABLE				MULTIVARIABLE *				PROPENSITY SCORE **			
	HR	Lower CI	Upper CI	*p*	HR	Lower CI	Upper CI	*p*	HR	Lower CI	Upper CI	*p*
All patients												
all	1.234	1.046	1.447	0.013	1.103	0.926	1.314	0.271	1.171	0.974	1.408	0.093
<pT2	0.830	0.440	1.567	0.566	0.844	0.430	1.659	0.624	0.777	0.401	1.505	0.454
≥pT2	1.218	1.017	1.458	0.032	1.219	1.015	1.463	0.034	1.226	1.011	1.486	0.038
non-NAC												
all	1.244	1.034	1.496	0.021	1.162	0.963	1.402	0.118	1.168	0.958	1.425	0.125
<pT2	0.584	0.252	1.353	0.210	0.726	0.314	1.680	0.454	0.555	0.232	1.331	0.187
≥pT2	1.220	1.009	1.477	0.041	1.229	1.013	1.492	0.036	1.241	1.011	1.524	0.039
NAC												
all	1.151	0.706	1.877	0.572	1.228	0.732	2.059	0.437	1.228	0.732	2.059	0.437
<ypT2	1.683	0.596	4.748	0.325	1.852	0.603	5.690	0.282	1.852	0.603	5.690	0.282
≥ypT2	1.090	0.624	1.904	0.762	1.136	0.630	2.046	0.672	1.136	0.630	2.046	0.672

* Addjusted for age, gender, severity of comorbidities reflected by American Society of Anesthesiologists, pathological T stage, pathological N stage, tumor grade, neoadjuvant chemotherapy. ** Weighted for age, gender, severity of comorbidities reflected by American Society of Anesthesiologists. CI: confidenceinterval; HR: hazard ratio; NAC: neoadjuvant chemotherapy.

## Data Availability

Source data available at: https://osf.io/bsnt5.

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
