# Peer review of "Diminishing the Gender-Related Disparity in Survival among Chemotherapy Pre-Treated Patients after Radical Cystectomy—A Multicenter Observational Study"

_jcm, 2023, doi:10.3390/jcm12041260_

Round 1
Reviewer 1 Report
This is an interesting and well written paper and I have enjoyed reading it.
There are some points to be addressed.
In the introduction says that long term outcome after RC, but the median follow up was 23 months, and 5-y survival was calculated. I guess it would be more correct to refer it as mid-term outcome instead of long term.
Clarify if there were any criteria for NAC administration. (ie ECOG)
Table 1. As far as I know chi square cannot be used when a variable is 0 as in ASA and chemo regimen. If you used Fischer exact test, please add it to M y M section.
Table 1 add % to the numbers to easily understanding and compare between groups.
Table 1 LN and SD are not in the table but are described below, please delete.
Figure add number of patients at risk.
There is a small number of patients with NAC and female as the number of patients. Could it and the almost two years of median follow up explain the no difference among gender?
Please expand: These findings were validated using a competing risk regression model. Which variables were used?
Discussion
Please change to regular font in “Comorbidity Index, and smoking status, which may have influenced survival. Secondly, there 231 were significant disparities between participating centers regarding chemotherapy regimens, num- 232 ber of administered cycles, and response to neoadjuvant treatment”
Reviewer 2 Report
This paper describes the impact of gender on overall survival in patients undergoing radical cystectomy. The conclusion is that NAC can help reduce gender specificity disparities, but I have some concerns as follows,
Major
1. After NAC, multivariate analysis shows no significant difference in patients with pT2 or higher (is this ypT2 or higher?), but I am afraid that this result means that only highly aggressive tumors eliminated the gender gap. This criticism may indicate a similar meaning that, for example, “Diminishing the Gender Discrepancy in pN1 positive patients after Radical Cystectomy” or “Diminishing the Gender Discrepancy in pT4 positive patients after Radical Cystectomy” If the effect of NAC is to be studied, the clinical stages should be applied and analyzed.
2. In the NAC group, the HR is considerably higher (though not significantly) in patients with pT2 or less (here I guess ypT2 or less), meaning that NAC might increase, not decrease, sex differences.
3. Table 1 should be divided into NAC+ and NAC – since the patients’ characteristics in patients with or without NAC is important to evaluate the conclusion.
4. Figure 2 should show the number of patients at risk, given that the median follow-up period is only 21 and 23.9 months, even though the study started in 1991. The authors should indicate that the number of NAC patients followed up was sufficient for proper analysis.
5. Multivariate analysis of the no NAC group should include whether or not adjuvant chemotherapy was administered.
6. Even anonymous data, did the Institutional Review Bord really exempt the review?
Minor
1. It is necessary to state how many patients were administered IO drugs if there were any.
Reviewer 3 Report
General comment
The manuscript entitled “Diminishing the Gender Disparity in Chemotherapy Pre-Treated Patients after Radical Cystectomy” aims to assess the role of gender in survival between patients who received NAC and those who underwent upfront radical cystectomy. The manuscript is overall well-written and understandable. Albeit the limitations reported by the authors, the topic of the study is not very consistent but could enrich the scarce literature regarding this issue. A few corrections are required in order to improve the quality of the work and clarify some points.
ABSTRACT
13: I would avoid starting with “several studies”. Please revise.
INTRODUCTION
36: Report possible explanations for this difference between males and females.
50: This section of the introduction seems truncated. Please extend the discussion and provide a wrap-up for the aim of your study.
RESULTS
108: This part could be included in the first section of the paragraph, avoiding the repetition of table 1 in brackets
DISCUSSION
204: Regarding the role of immunotherapy in bladder cancer in men and women please see: 10.3390/cancers14102545 and 10.3390/ijms23031133
210: About the difference between men and women in bladder cancer survival outcomes, in addition to the argument of bladder thickness, it should be also added that the factors which provoke this delay are related to the presentation of hematuria in women as cystitis and other UTIs (10.1002/cncr.28416 and 10.1002/cncr.25310)
CONCLUSIONS
253: Add future perspectives and hints for further research
Reviewer 4 Report
The aim of the study was to evaluate NAC impact on gender specific survival outcomes. Topic is interesting even if there are several critical aspects which Authors should reviewed in order to improve overall quality of manuscript.
- Title shoudl be revised becuase in its current form seems most focused on differences between male and female on receiving NAC
- Primary and secondary outcomes should be reported clearly.
- Baseline clinical features reported significant differences between male and female in terms of age and pTstage, both characteristics which could play a significant impact on survival outcomes. Therefore, a propensity scored match analysis should be performed.
- Neadjuvant chemoterapy response and rated of understage should be reported for both cohorts
- Cox regression analysis should be performed to evaluate impact of gender on OS, and in general to evalutate predictors of OS, such as pTstage, NAC, gender, in order to assess which is the independent predictor
Round 2
Reviewer 1 Report
Congratulation to the authors for improving the manuscript
Reviewer 2 Report
Although some limitations still remain, the authors have made appropriate revisions to the manuscript.
Reviewer 3 Report
The authors improved the manuscript accordingly. No further corrections are required in my opinion.
Reviewer 4 Report
No further comments, Authors properly addressed reviewer's suggestions.